# State-of-the-Art and Upcoming Innovations in Pancreatic Cancer Care: A Step Forward to Precision Medicine

**DOI:** 10.3390/cancers15133423

**Published:** 2023-06-30

**Authors:** Tommaso Schepis, Sara Sofia De Lucia, Antonio Pellegrino, Angelo del Gaudio, Rossella Maresca, Gaetano Coppola, Michele Francesco Chiappetta, Antonio Gasbarrini, Francesco Franceschi, Marcello Candelli, Enrico Celestino Nista

**Affiliations:** 1Center for Diagnosis and Treatment of Digestive Diseases, CEMAD, Gastroenterology Department, Fondazione Policlinico Universitario A. Gemelli, IRCCS, 00168 Rome, Italy; tommaso.schepis01@icatt.it (T.S.); sarasofia.delucia@guest.policlinicogemelli.it (S.S.D.L.); antonio.pellegrino02@icatt.it (A.P.); angelo.delgaudio01@icatt.it (A.d.G.); rossella.maresca01@icatt.it (R.M.); gaetano.coppola01@icatt.it (G.C.); antonio.gasbarrini@policlinicogemelli.it (A.G.); 2Department of Translational Medicine and Surgery, School of Medicine, Catholic University of the Sacred Heart of Rome, 00168 Rome, Italy; 3Section of Gastroenterology and Hepatology, Promise, Policlinico Universitario Paolo Giaccone, 90127 Palermo, Italy; michelefrancesco.chiappetta@gmail.com; 4IBD-Unit, Department of Clinical and Experimental Medicine, University of Messina, 98122 Messina, Italy; 5Department of Emergency Anesthesiological and Reanimation Sciences, Fondazione Universitaria Policlinico Agostino Gemelli di Roma, Catholic University of the Sacred Heart of Rome, 00168 Rome, Italy; francesco.franceschi@policlinicogemelli.it (F.F.); marcello.candelli@policlinicogemelli.it (M.C.)

**Keywords:** pancreatic cancer, therapy, multidisciplinary approach

## Abstract

**Simple Summary:**

Pancreatic cancer remains a major therapeutic challenge despite medical advances. The incidence of pancreatic cancer is increasing, and this disease is associated with a high mortality and morbidity rate. The poor prognosis of pancreatic cancer can be attributed to several factors, including the difficulty of early diagnosis due to the lack of specific symptoms and biomarkers in the early stages, the aggressiveness of the disease, and its resistance to systemic therapies. However, recent advances in the field have led to promising new therapeutic strategies, and endoscopists have assumed a key role in the multidisciplinary management of this disease. The aim of this article is to provide a comprehensive literature review focused on examining existing treatments for various stages of pancreatic cancer, with an emphasis on new and innovative therapeutic approaches.

**Abstract:**

Pancreatic cancer remains a social and medical burden despite the tremendous advances that medicine has made in the last two decades. The incidence of pancreatic cancer is increasing, and it continues to be associated with high mortality and morbidity rates. The difficulty of early diagnosis (the lack of specific symptoms and biomarkers at early stages), the aggressiveness of the disease, and its resistance to systemic therapies are the main factors for the poor prognosis of pancreatic cancer. The only curative treatment for pancreatic cancer is surgery, but the vast majority of patients with pancreatic cancer have advanced disease at the time of diagnosis. Pancreatic surgery is among the most challenging surgical procedures, but recent improvements in surgical techniques, careful patient selection, and the availability of minimally invasive techniques (e.g., robotic surgery) have dramatically reduced the morbidity and mortality associated with pancreatic surgery. Patients who are not candidates for surgery may benefit from locoregional and systemic therapy. In some cases (e.g., patients for whom marginal resection is feasible), systemic therapy may be considered a bridge to surgery to allow downstaging of the cancer; in other cases (e.g., metastatic disease), systemic therapy is considered the standard approach with the goal of prolonging patient survival. The complexity of patients with pancreatic cancer requires a personalized and multidisciplinary approach to choose the best treatment for each clinical situation. The aim of this article is to provide a literature review of the available treatments for the different stages of pancreatic cancer.

## 1. Introduction

Pancreatic cancer is an increasing health problem in Western countries. Pancreatic ductal adenocarcinoma (PDAC), a malignant tumor of the exocrine pancreas, is the most common cancer of the pancreas. Most PDACs arise from precursor lesions called pancreatic intraepithelial neoplasia, in a stepwise process through the acquisition of genetic alterations that eventually lead to the development of the invasive cancer. A minority of PDAC arises from cystic neoplasms such as intraductal papillary mucinous neoplasms (IPMN) [1]. According to a study by GLOBOCAN, the number of new cases of PDAC in 2020 was 495,773, ranking fourteenth on the list of most common cancers. The incidence of PDAC varies from country to country, with the highest incidence rates in Europe and North America and the lowest in Africa and Central Asia [2]. The prognosis of PDAC is poor, with a five-year relative survival rate of 12% according to the recent cancer statistics [3]. Difficult early diagnosis (lack of specific symptoms and biomarkers at early stages), the aggressiveness of the disease, and resistance to systemic therapies are the main factors leading to the high mortality rate of PDAC [4]. Tobacco use, type 2 diabetes mellitus, obesity, chronic pancreatitis, and heavy alcohol use are established environmental risk factors for PDAC [5,6,7,8]. Thus, only 5–10% of all pancreatic cancers are due to inherited risk factors, with not all cases of familial pancreatic cancer attributable to a known gene. Peutz–Jeghers syndrome (due to a mutation in the oncosuppressor gene STK11), hereditary breast ovarian cancer syndrome (due to mutations in BRCA1 or BRCA2), familial atypical melanoma (as a result of germline mutations in CDKN2A), ataxia telangiectasia syndrome (mutations in the gene ATM, which is involved in the DNA damage response), Lynch syndrome (mutations in the genes MLH1, MSH2, MSH6, which are involved in DNA repair) are associated with an increased risk of this cancer. In addition, patients with hereditary pancreatitis associated with mutations in SPINK1 and PRSS1 have a 40% lifetime risk of PDAC. Relatives from families meeting the familial pancreatic cancer criteria (at least one pair of first-degree relatives with pancreatic cancer) need genetic counselling and pancreatic surveillance. However, the modality, frequency, and age to begin it are still not defined [9].

Only a minority of patients diagnosed with PDAC have resectable disease. In fact, PDAC patients often present only non-specific symptoms (epigastric or back pain, nausea, bloating, distention, or change in stool consistency) before the disease reaches an advanced stage, which delays diagnosis. The occurrence of symptoms may vary depending on the location of the pancreatic tumor. For example, a pancreatic tumor localized in the head of the pancreas may result in obstructive jaundice due to biliary stricture, whereas pancreatic tumor localized in the body and tail of the pancreas continues presenting non-specific symptoms such as abdominal and back pain, loss of appetite, and weight loss. The new onset or exacerbation of preexisting diabetes may also be a sign of PDAC, regardless of its location [10,11,12]. Multidetector CT angiography with a dual-phasic pancreatic protocol is the most important diagnostic technique that can be used for the diagnosis, staging, and treatment planning of PDAC. Abdominal MRI is an alternative diagnostic method with the advantage of clear definition of biliary and pancreatic ducts.

Endoscopic ultrasonography is usually performed as part of PDAC diagnosis to better define vascular and regional lymph node involvement and to obtain a cytologic or histologic specimen for a definitive histopathologic diagnosis [11,13,14]. Several serological biomarkers have been studied in the management of PDAC, but their role is still limited. Serum carbohydrate antigen 19-9 (Ca19-9) is the most commonly used biomarker, primarily to monitor response to treatment and identify patients with poor prognosis despite surgery, but its value as a screening or early diagnostic tool remains controversial [15,16]. Other blood-based biomarkers such as cell-free DNA, exosomes, and circulating tumor cells are currently being investigated for monitoring response to treatment and assessing resistance to therapy [17]. PDAC staging is formally based on the tumor node metastasis (TNM) system based on the eighth edition of the American Joint Committee on Cancer Staging Manual. For the treatment algorithm, PDAC can be divided into resectable PDAC, borderline resectable PDAC (BRPC), locally advanced PDAC (LAPC), and metastatic PDAC, based on the degree of tumor abutment (tumor–vessel contact 180° or less) and encasement (tumor–vessel contact greater than 180°) in the celiac artery (CA), superior mesenteric artery (SMA), common hepatic artery (CHA), superior mesenteric vein (SMV), and portal vein (PV), as well as the presence of distant metastases [18,19]. Patients without distant metastases and without evidence of vascular involvement should be classified as anatomically resectable PDAC [20]. The first definition of “borderline resectable” disease was proposed by the M. D. Anderson Cancer Center Pancreas Cancer Group. A borderline resectable PDAC is a localized cancer with one or more of the following characteristics: an interface between the tumor and SMV-PV that measures 180° or more of the circumference of the vein wall, a short-segment occlusion of the SMV-PV with normal vein above and below that is amenable to resection and vein reconstruction, a short-segment contact of any grade between the tumor and hepatic artery with normal artery proximal and distal that is amenable to reconstruction, and an interface between the tumor and SMA or celiac trunk that measures less than 180° of the circumference of the artery wall [21]. In 2017, the International Association of Pancreatology published the latest classification model, which expands the anatomic definition of borderline resectability to include biological risk and patient condition. Under the new definition, borderline resectability includes tumor abutment or invasion of the SMV/PV with bilateral stenosis or occlusion not extending beyond the inferior border of the duodenum, tumor contact with the SMA and/or CA of less than 180° without stenosis or deformity, or tumor abutment of the CHA without tumor contact with the hepatic artery and/or CA. Borderline resectability has also been defined by a serum CA 19-9 level of >500 IU/mL and/or positive regional lymph node metastases (biological definition) and by poor patient performance status (PS of 2 or more, conditional definition) [22]. Locally advanced, surgically unresectable tumors encase the adjacent arteries (celiac axis, SMA, or both) or occlude the SMV, PV, or SMPV to an extent that does not permit venous reconstruction [23].

## 2. Advances in Pancreatic Surgery

Surgery is considered the best curative treatment for PDAC, although the 5-year survival rate is still about 10% [24]. In 80% of cases, patients are considered unresectable at the time of diagnosis [25]. A multidisciplinary team is required to determine proper PDAC staging and resectability of pancreatic cancer [25] (Figure 1).

The primary goal of surgical treatment is to achieve R0 resection. Although some previous studies have not shown a significant benefit for overall survival, several recent papers have supported the prognostic role of tumor-free margins (>1 mm) compared with R1 resection [26,27]. In addition to tumor removal with free margins, surgery also includes resection of at least twelve lymph nodes for pathologic staging [28]. Over the years, the surgical approach has changed from procedures with high mortality rates and poor survival to relatively safe, albeit difficult, procedures (the mortality rate is currently about 5%) [29]. This marked decrease in mortality rate is in particular due to the experience of hospitals with high surgical volume (>20 resections per year) [29,30,31]. Surgery followed by adjuvant therapy is the standard treatment for resectable tumors with a median survival of 26 months [32]. In recent years, the multidisciplinary approach has significantly changed the outcomes and indications for surgical treatment [33]. To date, up to 60% of initially surgically untreatable disease can convert to resectable disease after neoadjuvant therapies [34]. Some sources even report an increased 3- and 5-year survival rate of up to 30% in patients treated with a multimodality approach [35]. Because the rate of distant recurrence remains high, several studies in recent years have examined the role of neoadjuvant therapy in operable tumors [36,37,38]. This type of treatment appears to be associated with a lower lymph node positivity rate, better overall survival, and a higher rate of negative resection margins [39,40].

### 2.1. Surgical Approaches

Various techniques can be used in patients with pancreatic cancer, and their use depends on the anatomical locations of the disease. The classical approach for the treatment of pancreatic head tumors is the Whipple procedure. In contrast, distal pancreatectomy, usually in combination with splenectomy, is the typical technique for tumors localized in the body and tail [25]. The Whipple procedure, also known as pancreaticoduodenectomy, involves not only resection of the proximal portion of the pancreas but also removal of other adjacent organs such as the duodenum and proximal jejunum, common bile duct, part of the stomach, and gallbladder [29]. This surgical procedure is associated with a high rate of complications. The most common complications are intraoperative bleeding, pancreatic fistulas (5–15%), and delayed gastric emptying (DGE, 20%) [41,42]. Over the years, several surgical techniques have been adapted to limit complications and improve outcomes [41]. For example, “pylorus-preserving” and “stomach-preserving” techniques may reduce the incidence of DGE [43]. The “artery-first” approach, on the other hand, is a method in which the superior mesenteric artery (SMA) is dissected before any other resection steps [44]. A 2020 meta-analysis showed that this approach reduces complications and intraoperative blood loss and improves R0 resection margins [45]. In addition, surgical techniques involving resection and reconstruction of venous vessels, such as mesentericoportal vein axis, have been developed in recent years to achieve curative surgery even for borderline resectable PDAC or locally advanced tumors [46]. In a recent systematic review, the authors found that overall survival and complication rates in the venous resection group were similar to those of standard surgery despite the significantly higher R1 resection and blood loss [47].

### 2.2. Laparoscopy and Robotics Are the New Advances in Pancreatic Surgery

Given the considerable complexity and risks of pancreatic surgery, interest in minimally invasive surgical procedures has increased substantially in recent years [42]. Historically, the first presentations of laparoscopic procedures date back to the 1990s [48]. Subsequently, several studies have been conducted over the years to determine the safety and clinical outcomes of the laparoscopic approach. In 2018, a randomized clinical trial comparing perioperative outcomes between laparoscopic pancreatoduodenectomy (LPD) and open technique (OPD) was published [49]. Its data showed a significant reduction (13.5 vs. 17 days, *p* = 0.024) in the length of hospital stay in the LPD group, with no differences in surgical performance with respect to resection margins [49]. Similarly, the LEOPARD study compared minimally invasive pancreatectomy (MIDP) with traditional open pancreatectomy in patients with left-sided pancreatic tumors [50]. The authors focused primarily on postoperative recovery time and found that it was shorter in patients treated with MIDP (4 vs. 6 days, *p* < 0.001), with less intraoperative blood loss [51]. Croomee and colleagues examined the oncologic outcomes of total LPD compared with the open method. They concluded that the length of hospital stay was significantly shorter in the laparoscopic group [51]. In addition, more patients (12% vs. 5%; *p* = 0.04) started adjuvant treatment late or not at all in the OPD cohort. No statistical differences in overall survival (*p* = 0.22) were observed, as previously described in the literature [51,52,53,54]. The first robotic pancreatectomy (RPD) was presented in 2003 by Melvin et al., but not for malignant disease [55]. In a large comparative study of 1028 subjects, the robotic approach was associated with better perioperative outcomes, particularly lower blood loss and lower rates of major complications compared with open surgery [56]. Lower complication rates and fewer positive margins have also been noted in other meta-analyzes, underscoring the safety and efficacy of robotic pancreatectomy [57,58]. Moreover, compared with the laparoscopic approach, robotic surgery showed a lower need for intraoperative transfusions and a shorter hospital stay, with no statistical reduction in complication rates or R0 resection [59,60,61]. In addition, robotic surgery appears to be more expensive than LDP but very limited evidence is available [62]. Oncologic outcomes were examined in the DIPLOMA study, a pan-European propensity score-matching trial [63]. It included approximately 1200 patients in whom MIDP (both laparoscopic and robotic) was compared with open pancreatectomy. In this study, R0 resection and blood loss rates were better in the MIDP group. There were no improvements in postoperative morbidity and mortality [63]. In conclusion, the utilization of RDP and laparoscopic distal pancreatectomy (LDP) offers short-term clinical advantages, such as decreased blood loss and shorter hospital stays, when compared to ODP [60]. Nonetheless, it is crucial to take into consideration the existence of various biases. Firstly, patients who underwent RDP and LDP had notably smaller tumor sizes in comparison to those who underwent ODP. Additionally, the studies referenced in this analysis were predominantly retrospective and the overall sample displayed a substantial level of heterogeneity [60]. Consequently, the existing evidence concerning the oncological adequacy and safety of minimally invasive procedures in comparison to open procedures remains inconclusive and further randomized controlled trials are needed to clarify the role of these new surgical techniques in clinical practice.

## 3. Locoregional Therapy in Unresectable Pancreatic Cancer

### 3.1. Brachytherapy

In recent decades, significant progress has been made in the treatment of pancreatic cancer, characterized by the improvement of therapeutic approaches and the implementation of multidisciplinary measures, which has led to a marked improvement in disease outcomes [64]. However, locally advanced pancreatic cancer (LAPC) remains a major challenge for clinicians, as curative surgery is not always possible in cases of severe pancreatic vascular involvement. There is no consensus in the medical community on the optimal treatment for unresectable, locally advanced pancreatic cancer (LAPC). Current therapeutic options include multi-agent therapy, single-agent therapy, chemoradiotherapy, or stereotactic body radiation therapy (SBRT) [65]. Despite the advancements in therapeutic research regarding LAPC, the current therapies for tumor downstaging remain constrained. Currently used radiation doses achieve complete pathologic remission in only 10% of cases, and higher doses are needed to improve these rates [65]. A meta-analysis has shown that approximately one-third of patients with borderline resectable pancreatic cancer are eligible for radical resection after neoadjuvant therapy and that postoperative survival rates are comparable to those of patients with resectable pancreatic cancer at presentation [66]. Therefore, clinical investigators have explored alternative adjunctive therapies to neoadjuvant radiotherapy and chemotherapy to increase patients’ chance of curative surgery while limiting systemic therapy-related side effects. Systemic therapies typically cause significant side effects that can affect the overall well-being of patients, whereas locoregional therapies have gained attention due to their localized effect on the pancreas and can mitigate some of the barriers to effective antitumor treatment due to the tumoral microenvironment [67]. Brachytherapy is a targeted therapy for people with pancreatic cancer that provides localized treatment of the disease [68]. Radioactive seeds, microparticles, or fluids are delivered directly into the tumor to irradiate the neoplasm with a substantial dose of radiation without damaging healthy adjacent tissue. The use of iodine-125 (125-I) seeds as a means of brachytherapy has been widely used in the treatment of pancreatic ductal adenocarcinoma [69]. According to Gai et al., implantation of radioactive iodine-125 (125-I) seeds (RIS) is indicated for the treatment of pancreatic cancer in patients with a prognosis of more than 3 months who are not eligible for surgery or are ineligible because of their underlying disease; in patients with metastatic pancreatic cancer or local nodal metastases; in patients with residual cancer after resection of a pancreatic tumor; in patients with pancreatic tumor in difficult-to-reach regions; in patients with an expected survival time of less than 3 months as a possible alternative for relief of upper abdominal and back pain [70]. The seeds contain radioactive material and can be delivered into the tumor with an implantation needle and gun. Numerous methods are available for placing radioactive 125-I seeds into pancreatic tissue, including laparotomy or open surgery or percutaneous seed implantation, which is usually performed using CT scans or ultrasound guidance techniques [71]. Image-guided percutaneous puncture or endoscopic ultrasound-guided RIS implantation may be appropriate for patients with pancreatic cancer who are not candidates for resection. The percutaneous approach has several advantages over surgical techniques: precise localization, minimal trauma, significantly shorter operative time, faster postoperative recovery, less blood loss, and no abdominal scarring [71]. However, in patients requiring a gastrointestinal bypass, RIS implantation can be performed during surgery [70]. Intraoperative radiotherapy may also be chosen if inoperable tumors are discovered during laparotomy or if intraoperative evaluation indicates narrow or positive margins [72]. The use of radioactive 125-I seeds in conjunction with endoscopic drainage directly affects the containment of tumor proliferation in patients with pancreatic cancer and is therefore associated with longer survival, with a higher median survival time compared with control patients who underwent stenting alone [73]. In addition, a statistically significant increase in median duration of stent patency and median time to gastric outlet obstruction was observed in patients who underwent seed implantation compared with the control group. In addition, Xu et al. showed in their systematic review that intraluminal brachytherapy (ILBT) in patients with malignant obstructive jaundice resulted in a statistically significant reduction in the risk of stent occlusion and an improvement in median survival compared with stent implantation as the sole strategy [74]. Naidu and colleagues described the use of endoscopic ultrasound-guided intratumoral delivery of 32-phosphorus in conjunction with conventional cytotoxic therapy for the treatment of locally advanced pancreatic cancer (LAPC). The active implantable medical device contains P-32 in inactive silicon particles and directly damages cancer cell DNA, inhibiting further cell division and proliferation. This procedure proved to be practical and low in toxicity, reducing tumor size in 50% of cases and allowing surgical resection in 42% of patients [75]. The OncoPac-1 trial, a prospective, multicenter, single-arm pilot study, was conducted to evaluate the safety and efficacy of P-32 when implanted directly into the pancreatic tumor under endoscopic ultrasound (EUS) guidance in patients with unresectable LAPC undergoing chemotherapy. This resulted in a reduction in tumor volume from baseline and a significant decrease in CA 19-9 levels [65]. The most reported adverse events of brachytherapy included pyrexia (37.18%), abdominal discomfort (33.33%), vomiting (11.54%), and bowel irregularity (11.54%), while the most commonly observed adverse outcomes were pancreatitis (11.54%), infection (6.41%), bowel obstruction (1.28%), perforation of the digestive tract (1.28%), and seed migration (2.56%) through blood vessels and the pancreatic duct [76].

### 3.2. Ablative Therapies

Recently, several local ablation methods have been proposed for the mitigation of symptoms, inhibition of disease progression, and enhancement of survival rates in the treatment of inoperable pancreatic cancer with the intention to induce local tumor destruction [77]. Moreover, the development of ablative therapies for locally advanced pancreatic cancer (LAPC) has been influenced by the belief that such therapy, if applied with appropriate timing and indications, could enhance the likelihood of long-term survival for LAPC patients who are not candidates for surgical intervention. Techniques such as radiofrequency ablation (RFA), irreversible electroporation (IRE), high-intensity focused ultrasound (HIFU) have emerged as potential effective approaches in the handling of LAPC [78]. Radiofrequency ablation (RFA) is a thermal ablation modality used to achieve local tumor destruction by inducing high-frequency alternating current through the introduction of one or more intratumoral implanted electrodes [78]. Moreover, the application of RFA may have potential to serve as an immunomodulatory therapeutic intervention in patients with LAPC. Thus, Giardino et al. demonstrated the manifestation of RFA-induced immunomodulation in LAPC, by highlighting the systemic reaction elicited [79]. The response was characterized by a heightened adaptive immune response, with concomitant elevation of CD4+, CD8+, TEM, and myeloid dendritic cells activity, and a concurrent suppression of immune response. Furthermore, an extended activation of cells weeks post-procedure was observed, indicating a genuine immunomodulatory response rather than a typical inflammatory response. Figerio et al. conducted a randomized controlled trial (RCT) to assess the effects of laparotomic radiofrequency ablation (RFA) on multimodal therapy for locally advanced pancreatic cancer (LAPC) [78]. Notably, RFA monotherapy, as upfront therapy, exhibited no significant benefits over concurrent chemoradiotherapy (CHRT) regarding overall survival (OS) or progression-free survival (PFS). In addition, the CHRT group exhibited a greater prevalence of surgical resection following down-staging, if compared to the RFA group (21% vs. 4%). However, there was a comparable incidence of recurrence between the two groups, indicating that the therapeutic intervention did not impact the disease progression and further studies are necessary to assess the role of RFA. As a matter of fact, The PELICAN trial, an international multicenter ongoing randomized controlled superiority trial, aims to evaluate the incremental benefit of radiofrequency ablation (RFA) when administered concomitantly with chemotherapy monotherapy in patients with locally advanced pancreatic cancer in individuals exhibiting stable disease or partial response after two months of chemotherapy (CHT) intervention [80]. Paiella et al. conducted an analysis of the correlation between the genetic pathway specific to pancreatic ductal adenocarcinoma (PDAC), particularly the presence of SMAD4 mutation, and ablative procedures [81]. The mutation in the SMAD4 gene was observed in 60% of the analyzed patients. The utilization of SMAD4 analysis has been posited as a prospective mean of selecting a more suitable treatment for patients with PDAC. SMAD4 is a gene that is mutated in PDAC and is deemed to be one of the genes that plays a significant role in the promotion and progression of tumors. Specifically, this gene’s status has a direct impact on the tumor growth pattern, whether it be dominantly local or conducive to metastasis. More specifically, when SMAD4 is functioning with wild-type status, tumors exhibit an aggressive local manifestation, whereas mutations in the gene provide cancer cells with the ability to spread [81]. The assessment of SMAD4 status demonstrated the ability to differentiate LAPC sufferers who are prone to local progression, warranting the implementation of an ablation strategy. Due to the observed ability of RFA to induce tumor cytoreduction and potential immune stimulation, it may prove more effective in treating locally advanced pancreatic cancer cases where there exists a local predominant growth. Furthermore, Paiella et al. demonstrated that patients exhibiting preserved SMAD4 protein expression exhibit a stronger response to RFA, while those with an SMAD4 deficiency present a poorer prognosis in terms of survival if treated with RFA [81]. Currently, the analysis of SMAD 4 mutation in clinical practice for guiding pancreatic cancer ablative therapy has not yet been established.

Finally, Kumar et al. have delineated additional prospective applications of ablation, specifically the combined employment of radiofrequency ablation (RFA) and resection for increasing the likelihood of achieving R0 and R1 resections, leading to more advantageous clinical consequences for patients [82]. High-intensity focused ultrasound (HIFU) is an emerging non-invasive modality of pancreatic cancer ablative treatment which involves the bundling of high-intensity ultrasound waves using specialized transducers and concentrating them on a specific spot, resulting in a localized thermal ablation of the tumor with tissue destruction and coagulation necrosis in the targeted area. HIFU utilizes the collective effects of thermal, mechanical, and cavitation energies to induce instant coagulative necrosis of cancerous cells in the targeted area [77]. High-intensity focused ultrasound (HIFU) is an extracorporeal ablation modality that preserves skin integrity, obviating the need for surgical intervention or instrument insertion. Indeed, this ablative technique obviates the requirement of needles, electrodes, or probes, limiting any potential seeding of malignant cells into healthy adjacent tissues with negligible risk of puncture-associated hemorrhage. However, cutaneous burns may ensue in a small percentage of cases—between 0.4–1% [77]. Other side effects described were pancreatitis and pseudocyst collection [83]. It has been demonstrated that the use of preoperative high-intensity focused ultrasound (HIFU) ablation in borderline resectable pancreatic cancer, due to regional intravascular infiltration, has shown substantial improvements R0 resection rate, as well as mitigating challenges and potential risks associated with surgery [84]. Hence, HIFU may potentially serve as a valuable supplemental therapy for borderline resectable pancreatic cancer as a viable and efficient substitute to neoadjuvant chemotherapy (CHT) [84]. Results in terms of overall survival for HIFU have been demonstrated by Fergadi et al. wherein patients treated only with HIFU exhibited a significantly increased median overall survival (OS) compared to the chemotherapy-alone group [85]. Additionally, patients who were administered HIFU in conjunction with chemotherapy demonstrated higher OS at 6 and 12 months, when compared to those who received only standard chemotherapy. The co-administration of HIFU and chemotherapy may confer safety and efficacy for the treatment of advanced pancreatic cancer after gemcitabine failure [86]. A study conducted to determine the impact of HIFU and S-1 combination therapy on the prognosis of GEM-refractory metastatic pancreatic cancer demonstrated that the overall survival for the HIFU and S-1 combination therapy was 10.3 months compared to 6.6 months for S-1 monotherapy [86]. Additionally, it has been described that HIFU treatment results in sustained pain relief in 85% of patients with PDAC, with 50% of patients not necessitating analgesic therapy 6 weeks following ablation [77]. This analgesic effect was unaffected by the status of metastasis. The ablation of local nociceptive nerve fibers in the region contributes to the reduction of pain sensitivity due to the subsequent decrease in central nociceptive sensitivity. Additionally, the reduction of tumor size induces the decompression of surrounding structures, leading to further pain reduction [77]. Indeed, HIFU is a valid and safe alternative to neurolytic celiac plexus blockade (NCPB), which currently is considered the principal treatment modality offered in case of opioid therapy failure [85]. NCPB is an invasive procedure characterized by temporal efficacy, extending up to three months, and is also linked to significant undesirable side-effects such as localized pain, diarrhea, hypotension, pneumothorax, and neurological manifestations [85]. The limits of HIFU treatment are the need to visualize the target neoplasm with ultrasonic imaging with a depth not more than 12 cm. Furthermore, the presence of calcifications or surgical clips within the intended region may provoke an unsafe scattering of the sound waves [85].

Irreversible electroporation (IRE) is an additional ablative therapy, specifically, a non-thermal ablative modality, that utilizes high-voltage direct currents with short pulses of intense electrical fields [77]. This method prompts the formation of nanopores within cellular membranes, subsequently inducing permanent cell death [77]. IRE is achieved through the placement of two or more electrodes around the neoplastic tissue and can be performed percutaneously or intraoperatively [87]. Most commonly, IRE is conducted during a surgical procedure whereby the electrodes are positioned within the designated lesion, when the tumor is discovered to be non-resectable during the surgical procedure [77]. Aside from its palliative function in diminishing tumor mass, this approach may also be employed to reduce tumor stage via subsequent surgical intervention [77]. Oikonomou et al. provided evidence that the combination therapy involving IRE and chemotherapy for LAPC exhibited promising outcomes, as patients reported a median OS of 24.2 with a range of 6 to 36 months [88]. This combination therapy has been proven to be both successful and safe and may lead to greater survival outcomes. The two major complications observed were pancreatic fistula and delayed gastric emptying [88]. Additionally, a minor complication of wound infection occurred. Moreover, IRE may be employed as an adjunctive modality in open surgical procedures following primary resection, to attain localized control in instances where R0 resection was not perceived to be feasible, in the absence of a significant elevation in surgical complications [89]. Simmerman et al. noted zero instances of local recurrence of pancreatic tumors among all patients, with a median follow-up period of 14 months [89].

In a prospective study, He et al. compared the overall survival between cohorts of patients who underwent IRE with or without systemic therapy [90]. Among the participants who received or not chemotherapy for LAPC, no significant survival advantages were observed. These findings suggest that IRE played a crucial role in enhancing survival. Furthermore, the rates of overall survival (OS) and progression free survival (PFS) were greater for patients who received an induction of chemotherapy followed by IRE treatment than for those who underwent chemotherapy treatment alone [90]. Furthermore, it is increasingly gaining traction the concept of electrochemotherapy, which combines the use of IRE and hydrophobic regimens like bleomycin, able to increase the concentration of the latter within the tumor and enhance the effectiveness of treatment [91]. In their study, Rudno-Rudzińska et al. discovered compelling results in terms of overall survival in patients who underwent IRE + CHT [92]. Considering the destabilizing effect of IRE on the cell membrane, the combined use of with chemotherapeutic agent elevates the cytotoxicity of the drug while decreasing the required dosage, thereby reducing its overall toxicity [92]. Regrettably, the population under study was too heterogeneous and the sample size was limited. Indeed, a more extensive patient pool is necessary to establish conclusive statements. Ma et al. conducted a retrospective study to assess the effectiveness of combining percutaneous irreversible electroporation (IRE) with gemcitabine in comparison to IRE alone [93]. The Gemcitabine-IRE group exhibited a statistically significant increase in the median overall survival duration from the time of diagnosis, compared to the IRE group (21.5 months vs. 16.7 months, respectively). These outcomes suggest that the simultaneous utilization of gemcitabine and IRE represents a potent therapeutic strategy for patients diagnosed with LAPC [93]. The implementation of IRE in patients with LAPC who received FOLFIRINOX chemotherapy also appears to provide a potential improvement in overall survival (OS). Median OS was found to be significantly longer: 17.2 months, in patients who underwent IRE in addition to FOLFIRINOX chemotherapy compared to those who received FOLFIRINOX alone, with a median OS of 12.4 months [94]. The use of FOLFIRINOX chemotherapy has resulted in a positive impact on resection rates and survival trends for patients who have undergone induction chemotherapy for LAPC. However, in many cases, LAPC remains unresectable despite induction chemotherapy, and therefore, a combination of effective local ablation and systemic therapy may present a valuable approach for this group of patients [94]. A systematic review summarized the effect of IRE on local advancement of pancreatic cancer response, revealing that complete tumor remission was observed in 16% of patients, whereas partial tumor response was attained in 38.2% of patients [95]. Stable disease was reported in about 46.6% of patients, while 17.2% showed signs of disease progression. Moreover, 5.3% of patients were downstaged post IRE which eventually led to curative-intent surgery [95]. The mortality rate described was 4.4% for patients who underwent surgical IRE, while patients who underwent percutaneous or laparoscopic IRE did not report any instances of periprocedural mortality [95]. However, the current literature fails to completely elucidate the survival advantages of IRE for LAPC, and hence, unequivocal, and decisive proof supporting a survival benefit of IRE is not available. Regarding survival outcomes, local ablation methods may confer additional benefits to advanced pancreatic cancer patients, albeit with unproven efficacy. It seems that patients undergoing RFA and IRE display a comparatively higher median survival rate in contrast to those treated with HIFU [77]. This can be attributed to the selective utilization of RFA and IRE for patients with locally advanced disease lacking in distal metastases. In contrast, HIFU was applied also in individuals with advanced tumor stages and distal metastases [77]. To conclude, ablative techniques such as RFA, HIFU, and IRE necessitate further standardization and comparative studies before they can be assertively implemented for pancreatic cancer (PC) treatment in clinical settings. The favorable impact on overall survival (OS) demonstrated by these techniques in several studies, however, offers optimism for the efficacy of these therapies.

### 3.3. EUS-Guided Fiducial Marker Implantation and Intratumoral Delivery of Chemotherapeutic Agents

Pancreatic ductal adenocarcinomas have a hypovascular phenotype and dense fibrotic stroma, leading to suboptimal intratumoral delivery of systemic chemotherapy [96]. Therefore, EUS-guided peritumoral delivery of antitumor agents has emerged as a promising therapeutic approach for treating pancreatic cancer. EUS provides the ability to visualize the pancreas in real time and offers the unique advantage of direct intratumoral injection via the fibrotic stroma with improved regional concentration, thereby reducing systemic toxicity. EUS-guided fine-needle injection (EUS-FNI) with gemcitabine is a feasible method that has shown encouraging results in patients with pancreatic cancer (PC) for whom resection is not possible due to factors such as inoperability, poor operability, or patient refusal of surgery [97]. However, these results have yet to be validated by large-scale randomized clinical trials. In their phase I/IIa study, Fujisawa et al. investigated the efficacy of STNM01 as second-line therapy for gemcitabine plus Nab-paclitaxel-refractory, unresectable PDAC [98]. They observed that high tumoral expression of carbohydrate sulfotransferase 15 (CHST15) was associated with lower infiltration of CD3+ and CD8+ T cells at baseline. CHST15 is a proteoglycan-synthesizing enzyme and plays a critical role in remodeling stroma [99]. The authors administered an EUS-guided intratumoral injection of STNM01, a synthetic double-stranded RNA oligonucleotide against CHST15, resulting in significant suppression of tumoral CHST15 and a rapid induction of tumoral T-cell accumulation and prolongation of overall survival, which correlated significantly with the degree of increase in tumor-infiltrating CD3+ T cells at week 4 [98]. The results of this study demonstrate the ability of locally administered, EUS-directed RNA oligonucleotides to suppress tumoral CHST15 expression, thereby increasing the population of tumor-infiltrating T lymphocytes and ultimately optimizing patient prognosis [98]. EUS-FNI of the STNM01 oligonucleotide in patients with pancreatic cancer has demonstrated technical safety and successful tolerability [100]. However, it would be premature to draw conclusions and validate the clinical success of this therapy in terms of tumor size reduction, potential improvement in overall survival, and reduction in disease recurrence because randomization to a matched control cohort has not been performed and the effect of EUS-FNI on survival cannot be determined. Implantation of fiducial markers also represents a prospective approach to radiotherapy in the treatment of pancreatic cancer [101]. Fiducials are metallic or liquid radiopaque markers that are placed near or within target lesions and serve as internal landmarks that allow real-time tracking of the lesion. These markers serve as reference points for image-guided radiotherapy to deliver high-dose radiation with greater accuracy by quantifying respiratory motion and tumor extent and ultimately limiting the exposure of surrounding healthy tissue [102]. Originally, fiducial markers were introduced surgically or percutaneously using a CT scan [101]. However, these methods are invasive and carry a high risk of injury to adjacent organs and vessels. Endoscopic ultrasound (EUS) offers a less invasive approach and the possibility of high-resolution visualization of deep abdominal and mediastinal structures, avoiding the limitations of percutaneous insertion [103]. For pancreatic neoplasms, it is strongly recommended that endoscopic ultrasound (EUS) be used to place no fewer than three fiducial markers. Optimal placement of these markers targets the area of the tumor and/or its peripheral surroundings, ideally spreading them across different EUS levels and accurately marking the margin and planes of the tumor [101]. This targeted approach can lead to more effective local control rates while minimizing toxicity and it is gaining popularity as a means to improve tracking and localization during pancreatic radiotherapy, particularly in cases where onboard imaging systems lack the resolution to properly identify tumor–gut interfaces [101]. After the placement of EUS fiducial markers, CyberKnife treatment, a stereotactic body radiotherapy (SBRT) system, can be performed to deliver high doses of radiation to the target area. This treatment can be performed with the Synchrony motion-tracking module, which allows the tracking of tumors that exhibit respiratory movements [103]. Thus, EUS-SBRT with the CyberKnife module is a successful and targeted approach that spares healthy tissue in patients for whom conventional radiotherapy is not an option [103]. The technical feasibility of EUS-guided fiducial placement was demonstrated with an impressive overall success rate of 98%. Spontaneous migration of the fiducial may rarely occur. Bhutan et al. noted an overall migration rate of only 3%, with no complications [101]. The use of stereotactic body radiotherapy (SBRT), in contrast to conventional radiotherapy, has been associated with better local control and longer survival because higher doses of radiation can be administered, resulting in better outcomes with an acceptable toxicity profile [101]. However, Moningi et al. reported that fiducial placement had no effect on surgical outcomes, overall survival, and local recurrence [104]. The lack of a beneficial effect of fiducial placement may be attributed to the higher proportion of patients with locally advanced pancreatic cancer in the fiducials group (46.0%) compared with the control group (30.4%) [104]. In addition, the radiation doses administered to the two groups did not differ significantly, with a median dose of 38 Gy in the fiducial group versus 36 Gy in the control group. These results should reemphasize to clinicians that high-quality, image-guided radiation therapy (IGRT), with or without fiducials, is critical to stereotactic body radiation therapy (SBRT) [104]. Thus, the actual benefit of fiducial marker placement is still under investigation; however, endoscopists are becoming important players in the multidisciplinary team treating patients with PDAC.

## 4. Chemotherapy

In the early stages of PDAC, the current approach is surgical resection followed by adjuvant therapy. However, the beginning of a neoadjuvant therapy protocol before surgery remains controversial [105].

The American Society of Clinical Oncology (ASCO) guidelines recommend limiting neoadjuvant therapy under certain conditions, such as suspected extrapancreatic disease, radiologic description of mesenteric vessel infiltration, or increased surgical risk [106]. However, in the recent PREOPANC trial, neoadjuvant therapy based on gemcitabine and radiotherapy followed by surgery and adjuvant therapy were shown to be superior to prior surgery followed by adjuvant therapy in both resectable and borderline resectable pancreatic cancer [107]. In addition, a large randomized phase III trial comparing perioperative versus adjuvant chemotherapy in resectable disease is currently ongoing (NCT04340141). In borderline resectable and locally advanced disease, surgery is not recommended as the first step by the National Comprehensive Cancer Network (NCCN) [20]. Therefore, at this stage, the choice is between chemotherapy or chemoradiotherapy as neoadjuvant therapy, and FOLFIRINOX (a combination of folinic acid, fluorouracil, irinotecan, and oxaliplatin) as well as gemcitabine-cisplatin and gemcitabine-nab-paclitaxel are currently accepted treatment options [106]. The choice and modalities of adjuvant therapy are unclear. Interestingly, recent studies have examined the role of biomarkers, such as circulating tumor DNA, in guiding the intensity of adjuvant therapy, with encouraging results [108]. However, in this situation, modified FOLFIRINOX (mFOLFIRINOX), based on reduced irinotecan and without 5 FU bolus, has been shown to be superior to gemcitabine treatment in the PRODIGE-24 trial, although it is mainly used in patients with good performance status [109]. In cases where the above chemotherapy regimen is contraindicated, the gemcitabine/capecitabine combination (GemCap) is usually employed. This combination is superior to gemcitabine monotherapy [20]. However, in Asia, adjuvant therapy is usually based on a combination of tegafur (5 FU prodrug), potassium otteracil, and gimeracil, which was considered non-inferior to gemcitabine in the JASPAC-01 trial [110]. Unfortunately, most patients in this trial presented advanced or metastatic disease, and the median one-year survival rate was 7% [111]. Therapeutic strategies for metastatic disease are based on the overall status of the patient or the identification of specific genetic mutations [18]. These include FOLFIRINOX, gemcitabine and capecitabine, gemcitabine and cisplatin, nab-paclitaxel plus gemcitabine, or gemcitabine and erlotinib. Other treatments at this stage include FOLFOX (the combination of 5 FU, oxaliplatin and leucovorin) or FOLFIRI (combination of 5 FU, folic acid and irinotecan). In patients with low performance status, therapy based on gemcitabine alone can also be used [105]. Interestingly, the development of next-generation sequencing and bioinformatics techniques has identified numerous genetic mutations implicated in the primary mechanisms of carcinogenesis and responsible for a distinct subset of PDAC [112]. Some of these mechanisms are illustrated in Figure 2.

These changes affect different phases of the cell cycle, e.g., activation of genes that stimulate proliferation, gene transcription processes, or DNA repair mechanisms, and may result in a differential response to therapies [112]. For example, pancreatic ductal adenocarcinoma (PDAC) characterized by microsatellite instability (MSI) (constituting approximately 1% of PDAC cases) exhibits diminished responsiveness to 5-fluorouracil (5-FU) and gemcitabine treatments, while also presenting a better response to FOLFIRINOX therapy and immunotherapy [111]. Notably, a mechanism of increasing importance is the loss of high-fidelity homologous double-strand break (HR) recombination. Recent studies involving individuals with breast cancer susceptibility protein (BRCA1/2 genes) mutations, detected in approximately 10% of pancreatic ductal adenocarcinomas (PDACs), have revealed analogous characteristics to breast cancer cases associated with such mutations. Notably, these PDAC cases exhibit a more favorable therapeutic response to interventions that provoke double-stranded DNA breaks, such as platinum salts. Accordingly, increased response rates to FOLFIRINOX therapy have been reported in these patients [113,114,115]. Some authors emphasize the importance of germline and somatic testing at diagnosis to predict a better response to treatment [116,117]. In addition, poly ADP ribose polymerase (PARP) inhibitors such as niraparib and olaparib also show promising results in this group of patients. These drugs, which block the DNA-reparative action of PARP proteins, could lead to cell death, preferably in cancer cells [118]. Similar considerations also include the presence of mutations in the Partner and localizer of the BRCA2 (PALB2) gene. Accordingly, Rucaparib, a PARP inhibitor, has demonstrated promising results in the context of BRCA/PALB2-mutated pancreatic ductal adenocarcinoma (PDAC). Currently, a phase 2 clinical trial is underway to investigate the efficacy of Rucaparib as a single-agent therapy in patients with germline or somatic mutations in BRCA1, BRCA2, or PALB2 genes. In addition, Olaparib was recently evaluated and approved as a maintenance therapy in patients with PDAC, as evidenced in a phase 3 study (POLO), specifically for BRCA-mutated patients who had not advanced during first-line platinum-based chemotherapy.

Unfortunately, resistance to PARPi is high due to mechanisms such as upregulation of drug efflux pumps or changes in the tumor microenvironment [119]. Further genomic studies have revealed the presence of nonclassical genes that may be linked to homologous recombination deficiency (HRD) and could potentially affect the response PARPi. A comprehensive meta-analysis examined various surrogate markers of HRD in PDAC, with a specific focus on BRCA1, BRCA2, PALB2, ATM, ATR, CHEK2, RAD51, and FANC [120]. For instance, ATR (Ataxia Telangiectasia and Rad3-related protein) plays an important role in regulating cell cycle checkpoints and homologous recombination processes [121]. Some data suggest that inhibition of PARP leads to an increased reliance on ATR/CHK1 checkpoint signaling [122]. Consequently, multiple studies support the combination of ATR and PARP inhibition. In breast cancer cells, the combination of ATR inhibition and PARP inhibitors has a synergistic effect in promoting cell death, regardless of homologous recombination deficiency [123]. Additionally, ATR inhibition by VE-821 overcomes resistance to PARP inhibitors in cells lacking BRCA function [123].

Furthermore, various inhibitors of cell cycle regulators are currently being studied in combination with PARP inhibitors, both in tumors lacking homologous recombination capability and those with intact mechanisms.

Notably, the most frequently observed mutations in PDAC involve the Kirsten rat sarcoma viral oncogene homolog gene (KRAS); the G12D and G12V mutations occur in approximately 90% of cases and are specific to pancreatic cancer. Not surprisingly, such mutations play an important role in tumorigenesis by contributing to immune evasion, recruitment of suppressive immune cells, and metabolic alterations [124]. Interestingly, recent studies have found prognostic differences between the different mutations; patients with the KRAS -G12D subtype had shorter overall survival than patients with the KRAS wild type [125]. However, the introduction of specific gene inhibitors has already shown good results in other malignancies. In PDAC, promising results have been obtained with sotorasib and adagrasib, drugs targeting a rare (1−2% of PDAC) G12C mutation [126,127]. Interestingly, an ongoing phase I/II study (CodeBreak100- NCT03600883) is evaluating the safety and tolerability of sotorasib in adult subjects with KRAS p.G12C mutant advanced solid tumors. There have been reports of sotorasib anti-cancer effects demonstrated in 8 of 38 individuals with advanced pancreatic cancer who have previously received treatment and had KRAS p.G12C mutation [128]. Other approaches under development include pan KRAS inhibitors, specific G12D inhibitors such as the G12D inhibitor MRTX1133, and T-cell engineering targeting KRAS G12D [129]. In contrast, tumors without KRAS mutation seem to be more sensitive to epidermal growth factor receptor (EGFR) blockade, as shown by the NOTABLE phase III trial with the combination of gemcitabine plus nimotuzumab (an anti-EGFR monoclonal antibody) [130]. Other interesting therapeutic approaches involve tumor metabolic pathways, such as mitochondrial metabolism and ATP production. Although the combination of devimistat, an inhibitor of the tricarboxylic acid cycle in mitochondria, and modified FOLFIRINOX provided positive results in a phase I trial; the data were not confirmed in the phase III trial [131]. Relevant features currently under investigation involve the tumor stromal environment, which plays an important role in the mechanisms of therapy resistance and tumor progression. Strategies based on stromal modifiers such as pegvorhyaluronidase alpha (PEGPH20) are being tested in combination with mFOLFIRINOX [132]. Finally, as mentioned previously with olaparib, selected patient populations may benefit from maintenance therapies. Combinations of 5-FU and leucovorin or gemcitabine have been tested in clinical trials with promising results [133,134]. In addition, some studies are investigating the use of maintenance therapies in patients with minimal residual disease identifiable by novel markers such as ctDNA to prolong survival. Despite recent improvements in treatment, treatment resistance is one of the most important factors affecting the prognosis of pancreatic cancer [135,136]. The gut microbiota is not only involved in the pathogenesis of PDAC, but also seems to play an important role in therapy resistance, mainly by influencing drug metabolism and absorption [137,138]. Some bacteria belonging to the class of gamma-proteobacteria, such as Escherichia coli, are abundant in human PDAC and can affect gemcitabine metabolism by converting this drug to its inactive form (2′,2′-difluorodeoxycytidine to 2′,2′-difluorodeoxyuridine) via their long isoform of the enzyme cytidine deaminase (CDA). The same effect on gemcitabine metabolism has also been attributed to Mycoplasma spp. probably due to the same enzymatic pathway; indeed, CDA inhibitors such as tetrahydrouridine (THU) were found to be helpful in restoring drug activity in the same study [139]. These effects have been further explored in colorectal cancer, and interesting, albeit limited, data are currently available on pancreatic cancer as well. In addition, Weniger et al. reported that the presence of K. pneumoniae in bile ducts was associated with a worse outcome in a population treated with adjuvant gemcitabine therapy; in contrast, higher survival was observed after antibiotic therapy with quinolones [140]. Fluoropyrimidine therapies, which are part of the FOLFIRINOX regimen, also appear to be affected by microbiota-mediated chemoresistance [141]. Bronckaers et al. noted a reduction in the activity of pyrimidine nucleoside analogs in cell lines infected with Mycoplasma hyorhinis, which contains multiple nucleoside-metabolizing enzymes. In support of this causal relationship, drug efficacy was fully restored by the thymidine phosphorylase inhibitor TPI [141]. Another bacterium blamed for drug resistance is Fusobacterium nucleatum [142]. Interestingly, the abundance of F. nucleatum has been associated with poor prognosis in pancreatic cancer, and an increased rate of inactivation of 5-FU by this bacterium has been found in colon and rectal cancers. Mechanisms involved may include interaction with TLR4/MYD88-dependent autophagy and suppression of 5-FU-induced cell apoptosis [143,144]. In addition, F. nucleatum has been associated with oxaliplatin resistance [144], although no specific enzyme or mechanism is currently known. Other bacterial species appear to play a positive role in supporting chemotherapeutic activity; in particular, Bacteroides ovatus and Bacteroides xylanisolven have been associated with increased T-cell recruitment and enhanced effects of erlotinib, an EGFR tyrosine kinase inhibitor [145].

## 5. The Quality of Life

Pancreatic cancer can significantly affect a patient’s quality of life, both physically and emotionally. Symptoms such as pain, fatigue, nausea, and loss of appetite can interfere with a patient’s ability to perform daily activities, while psychological problems and anxiety can also have a significant impact on a patient’s well-being [146]. Treatment of the symptoms associated with pancreatic cancer is one of the first goals in the management of this disease. Indeed, survival is not the only outcome to be evaluated, but quality of life and the patient’s perception of the disease are cornerstones in the treatment of malignancies.

### 5.1. Pain Management

Pain is a common symptom of pancreatic cancer and occurs in up to 85% of patients [147]. It may be caused by tumor growth or inflammation, nerve involvement, or treatment-related side effects. Pain management in pancreatic cancer may include a combination of medications with other measures. Acetaminophen and nonsteroidal anti-inflammatory drugs (NSAIDs) are commonly used for pain management and are generally considered first-line agents [148,149]; however, these medications can only relieve mild to moderate pain in pancreatic cancer and may not be effective for severe pain. Opioids are the most effective drugs for treating mild to moderate pain that does not respond to first-line treatment or moderate to severe pain in pancreatic cancer [150]. Opioids can be administered orally, transdermally, or intravenously and should be tailored to the individual patient’s pain level. Tramadol, dihydrocodeine, and codeine are generally used to treat mild to moderate pain [151,152]. For severe pain, strong opioids such as morphine, oxycodone, or fentanyl may be used, but careful attention must be paid to side effects (e.g., constipation, nausea, vomiting, respiratory depression, cognitive impairment, confusion, hallucinations, opioid-induced hyperalgesia) [153]. Concomitant medications such as antidepressants, anticonvulsants, and corticosteroids can enhance the effects of opioids and improve pain control [150]. For example, tricyclic antidepressants (TCAs) may be effective in treating neuropathic pain, and gabapentin may be used to treat neuropathic pain and opioid-induced hyperalgesia. In patients with pain associated with bone metastases, standard pain management can be combined with bisphosphonates, external beam radiotherapy, and stereotactic body radiotherapy to improve analgesia [154,155]. Unfortunately, in up to 10% of patients, pain is refractory to analgesics. In this case, invasive pain management with procedures such as nerve or neurolytic blocks and intrathecal drug administration should be considered [156]. Endosonography-guided celiac plexus neurolysis (EUS-CPN) offers an alternative method for achieving short-term pain control in patients with non-surgical pancreatic cancer [157]. This technique offers the potential to reduce opioid dosages or be considered for uncontrolled pain with conventional opioid therapy. The effectiveness of EUS-CPN ranges from 50% to 94% across different studies, with a pain relief duration lasting between 4 and 8 weeks [157]. EUS-CPN is generally considered a well-tolerated technique, with reported complications accounting for approximately 44% of cases, although the majority of these incidents have been minor and transitory in nature [157]. Nonetheless, the current literature does not provide evidence of improved patient survival or enhanced quality of life following the implementation of EUS-CPN.

### 5.2. Nutrition

Nutritional assessment is an essential component of pancreatic cancer management. Adequate nutritional support is essential to maintain patients’ nutritional status, improve their quality of life, and potentially increase the effectiveness of treatment. The spectrum of malnutrition in pancreatic cancer can lead to cancer cachexia, which can be defined as a multifactorial syndrome leading to loss of skeletal muscle mass and progressive physical impairment [158]. Nutritional support in patients with pancreatic cancer should be initiated as soon as possible to prevent malnutrition and maintain the patients’ functional status. The nutrition plan should be individualized based on the patient’s nutritional status, disease stage, and treatment plan. A dietitian can help create a personalized nutrition plan based on the patient’s specific needs. Patients with pancreatic cancer should eat a diet high in calories and protein to maintain energy balance and preserve muscle mass [159]. Pancreatic enzyme replacement therapy (PERT) can be used in patients with pancreatic cancer who have pancreatic insufficiency and malabsorption. PERT can improve digestion and the absorption of nutrients and reduce symptoms of malabsorption, such as diarrhea and steatorrhea [160]. Oral supplements, enteral nutrition, or parenteral nutrition may be necessary to support patients’ nutritional status when they cannot meet their energy and nutrient needs through diet alone [161].

### 5.3. Psychological Assessment

Addressing the psychological needs of patients with pancreatic cancer is an important component of their care and can help improve their quality of life and treatment outcomes [162]. Pancreatic cancer is a difficult disease with a high mortality rate, and patients and their families may experience significant emotional distress, including depression, anxiety, fear of recurrence, and adjustment difficulties. These psychological factors can impact patients’ ability to cope with their disease, adhere to treatment plans, and make decisions about their care. Psychological interventions can help address these factors and improve patient outcomes. For example, cognitive behavioral therapy can be effective in treating depression and anxiety, while psychoeducation can help patients and families better understand their illness and treatment options [163]. Support groups, individual counseling, and peer support can be helpful in providing emotional support and reducing feelings of isolation and anxiety.

## 6. Conclusions

Pancreatic cancer is a significant health problem worldwide, with a high mortality rate and limited treatment options. According to the World Health Organization (WHO), pancreatic cancer is the seventh leading cause of cancer-related deaths worldwide. The five-year survival rate for pancreatic cancer is low, with only 10–20% of patients surviving five years after diagnosis [163]. The prognosis for pancreatic cancer is generally poor, and the disease is often diagnosed at an advanced stage, making treatment difficult. Treatment options for pancreatic cancer depend on the stage and location of the cancer, as well as other factors such as the patient’s overall health. A multidisciplinary team of professionals, including oncologists, surgeons, and radiation oncologists, can help determine the best treatment plan for each individual patient [163]. Today, the only way to survive pancreatic cancer is surgery. The type of surgery depends on the location and extent of the cancer. Pancreatic surgery is challenging with high mortality and morbidity rates, but with the advent of minimally invasive surgery and robotics, outcomes are steadily improving [46]. For pancreatic cancer, surgery is rarely the only treatment but is often combined with adjuvant chemotherapy or, in selected cases, neoadjuvant chemotherapy. In patients who are not candidates for surgery, chemotherapy is the main therapeutic approach [164]. FOLFIRINOX, gemcitabine in combination with capecitabine, cisplatin, or nab-paclitaxel are the most commonly used drugs for the treatment of pancreatic cancer. In selected cases, locoregional therapies may play a role in the management of locally advanced pancreatic cancer. Radiofrequency ablation (RFA), irreversible electroporation (IRE), high-intensity focused ultrasound (HIFU), EUS-guided implantation of fiducial markers, and administration of intratumoral agents have been described, but their role in daily practice remains to be clarified [65]. In addition to survival, quality of life is an important outcome in patients with pancreatic cancer. Management of pain, nutritional status, subjective symptoms (e.g., fatigue, nausea, vomiting), and the psychological needs of patients and their families are considered essential components of the medical management of pancreatic cancer.

To reduce the global burden of pancreatic cancer, prevention, early detection, and treatment options must be improved. Research into the causes of pancreatic cancer is also critical in order to develop more effective therapies and improve treatment outcomes and prevention strategies.

## Figures and Tables

**Figure 1 cancers-15-03423-f001:**
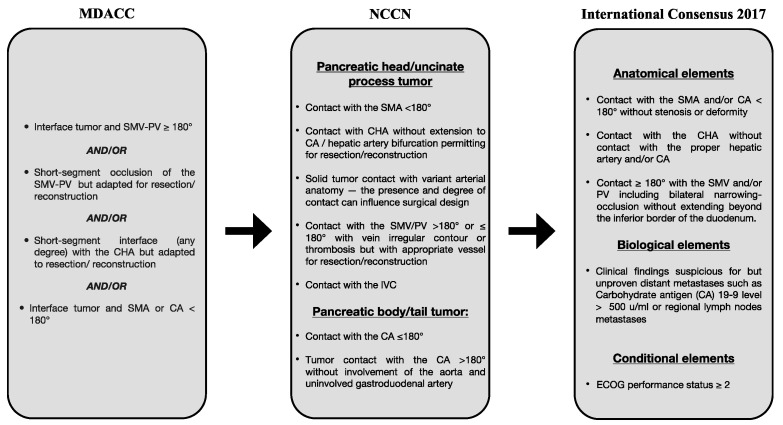
Resectability criteria for BR-PDAC. MDACC = MD Anderson Cancer Center; NCCN = National Comprehensive Cancer Network; SMA = superior mesenteric artery; SMV = superior mesenteric vein; IVC = inferior vena cava; PV = portal vein; CA = celiac axis; CHA = common hepatic artery; ECOG = Eastern Cooperative Oncology Group.

**Figure 2 cancers-15-03423-f002:**
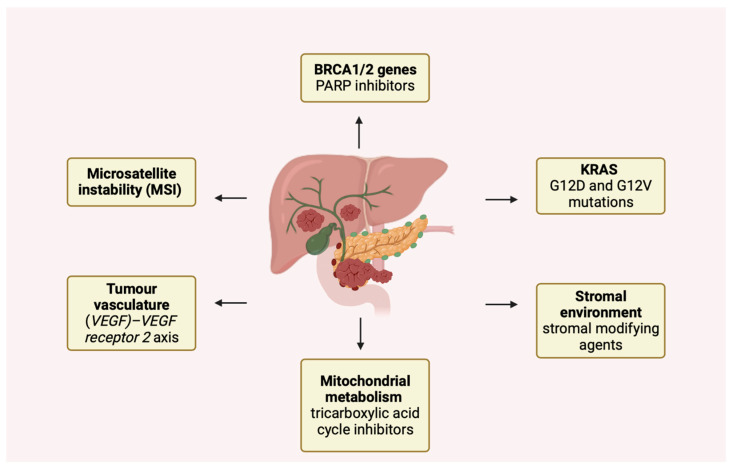
Novel molecular mechanisms involved in the pathogenesis of pancreatic cancer.

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
