# Peer review of "State-of-the-Art and Upcoming Innovations in Pancreatic Cancer Care: A Step Forward to Precision Medicine"

_cancers, 2023, doi:10.3390/cancers15133423_

Round 1

Reviewer 1 Report

The Authors write a review about pancreatic cancer, and include in title the idea of personlized medicine.

Minor observations or changes required: 

- line 188  consider changing "however" as the first word of the sentence 

- line 200 check for missprint

- line 210-211 "despite....." check the meaning of the sentence

-lines229-235 consider changing the content ina table; check for style, Capital or low case "I"

-method for seeds delivery can be moved and fitted into sentence in line 236

line 243 check missprint

lines 266-270 consider including some percentages of the complications

line 289 consider avoiding the use of suppression twice in close words

line 309 consider including the percentage of  cases withSMD4 mutations

line 313 consider specifying if germinal or somatic mutations are considered

line 345 "Patients" change in "patients"

line 362 consider changing/erasing the first word in the sentence "Though"

-line 381. ....of 24.2  >add months or appropriate time interval

-421 systematic >Systematic

line 428     extant ?

line 467  consider removing "Therefore"

-line 516 citation of figure 2 in this place, seems inappropriate

Major suggestions or changes requested:

 chapter 3.1

consider  writing it improving sequence of topics, mainly in the first part.;

lines 51 to 58 :  the authors correctly list genes whose grmline mutations may cause pancreatic cancer (PC);  They should add also the information about overall familial recurrence of PC.

The authors correctly discuss the issues about late diagnosis  (line 701 ) and early diagnosis (line 722); the should complete discussion including, even shortly, for cases with gemline mutations,  the issues of genetic testing for relatives, genetic counselling and follow up of asymptomatic carriers of  germline mutations.

English Language seems good; 

please consider checking the use of adverbs at the beginning of phrases, they often seem to me not well related to the meaning of the sentence.

Author Response

Dear Editor,

Many thanks for giving us the opportunity to resubmit our paper “STATE OF THE ART AND UPCOMING INNOVATIONS IN PANCREATIC CANCER CARE: A STEP FORWARD TO PRECISION MEDICINE”.

We also wish to thank the reviewers for their time and effort. We went through their interesting suggestions that allowed us to improve our paper.

Please, find a point-by-point answer to all addressed issues:

REVIEWER 1

Minor observations or changes required:

  1. line 188 consider changing "however" as the first word of the sentence. R: We thank the reviewer for this suggestion and we replaced “however” with the word “moreover” for a more fluid sentence
  2. line 200 check for misprint. R: We thank the reviewer for the comment and provided the correction “Locoregional Threapy” with “Locoregional Therapy”.
  3. line 210-211 "despite....." check the meaning of the sentence. R: We thank the reviewer for the comment and provided a clearer sentence as follows: Despite the advancements in therapeutic research regarding LAPC, the current therapies for tumor downstaging remains constrained.
  4. Lines 229-235 consider changing the content ina table; check for style, Capital or low case "I". R: We thank the reviewer for the comment and modified I125 with I-125.
  5. method for seeds delivery can be moved and fitted into sentence in line 236 R: we thank the reviewer for the suggestion, we modified the text as requested.
  6. line 243 check missprint. R: We thank the reviewer for correction and modified “scarring71” erasing “71”.
  7. lines 266-270 consider including some percentages of the complications. R: We thank the reviewer for the comment. As suggested, we described the percentages of the adverse events of brachytherapy: pyrexia (37.18%), abdominal discomfort (33.33%), vomiting (11.54%), and bowel irregularity (11.54%), pancreatitis (11.54%), infection (6.41%), bowel obstruction (1.28%), perforation of the digestive tract (1.28%), and seed migration (2.56%) through blood vessels and pancreatic duct.
  8. line 289 consider avoiding the use of suppression twice in close words. We thank the reviewer for correction and modified the text with “suppression of immune response”.
  9. line 309 consider including the percentage of cases with SMD4 mutations. R: We thank the reviewer for the comment. As suggested, we modified the text including the percentage of cases with SMAD4 mutation (60% of the analyzed patients).
  10. line 313 consider specifying if germinal or somatic mutations are considered. R: We thank the reviewer for the comment. Regrettably, the study does not provide explicit details regarding this information.
  11. line 345 "Patients" change in "patients". R: We thank the reviewer for the comment and modified as request.
  12. line 362 consider changing/erasing the first word in the sentence "Though". R: We thank the reviewer for the comment and modified as request.
  13. line 381. ....of 24.2  >add months or appropriate time interval. R: We thank the reviewer for the comment and modified as request adding the range (6 to 36 months).
  14. 421 systematic >Systematic. R: We thank the reviewer for the comment and modified as request.
  15. line 428     extant ? R: We thank the reviewer for the comment and replace the word “extant” with the word “current”.
  16. line 467  consider removing "Therefore". R: We thank the reviewer for the comment and replace the word “Therefore” with the word “Though”.
  17. line 516 citation of figure 2 in this place, seems inappropriate. R: We thank the reviewer for the comment and provided a new disposition of the figure in order to ensure a more fluid reading of the paragraph.

Major suggestions or changes requested:

consider writing it improving sequence of topics, mainly in the first part.;

  1. lines 51 to 58:  the authors correctly list genes whose germline mutations may cause pancreatic cancer (PC);  They should add also the information about overall familial recurrence of PC. R: We thank the reviewer for the comment and add the percentage of familiar recurrence of pancreatic cancer (5-10%) as suggested.

  1. The authors correctly discuss the issues about late diagnosis  (line 701 ) and early diagnosis (line 722); the should complete discussion including, even shortly, for cases with gemline mutations,  the issues of genetic testing for relatives, genetic counselling and follow up of asymptomatic carriers of  germline mutations. R: We thank the reviewer for the comment. We better displayed the indication of genetic counselling and pancreatic surveillance as suggested.

Reviewer 2 Report

I thank the authors for the opportunity to review their study, entitled: "PERSONALIZED MANAGEMENT OF PANCREATIC CANCER: A STEP FORWARD TO PRECISION MEDICINE"

Overall, the manuscript is very expansive and covers many of the basic and faceted aspects of pancreatic cancer care.

The manuscript itself could benefit from some English language editing: There are no major grammatical errors overall, but the order and organization of the sentences and paragraph could be improved to increase readability and clarity.

Specific comments:

1. Despite of the title, most of the manuscript does not touch upon personalized or precision medicine. In fact, I would estimate that only a page's worth discussed personalized medicine aspects. I would recommend to considers changing the title to something more fitting like "State of the art and upcoming innovations in pancreatic cancer care".

Introduction:

2. Consider citing more recent statistics (2023) for pancreatic cancer survival.

3. Line 59 - Should be a start of a new paragraph

4. Line 84 - Imaging TNM ?

5. Line 91 - Patients should be classified as anatomically resectable, other conditions may preclude them from surgery.

Advances in Pancreatic Surgery:

6. Line 113 - Surgery coupled with post/perioperative chemotherapy is the only way to cure...

7. Figure#1 - Font is too small, please enlarge.

8. Line 122 - R0 is not microscopic

9. Line 125 - R1 is not macroscopic

10. Line 130 - Recent large studies use 20 or even 35 cases/year as the threshold for high-volume centers.

11. Line 131 - Are the authors referring to minimally invasive surgery (laparoscopic/robotic)? if so, what are the RCTs or Meta-analyses supporting MIS having lower mortality?

12. Lines 170-199 - It may be worthwhile to cite some of the contrasting literature, as there is no clear consensus about the superiority of MIS in pancreatic surgery. Especially, since not all tumors are amenable to MIS resection and therefore there is in inherent selection bias in some of the studies.

Locoregional Therapy in Unresectable Pancreatic Cancer:

13. Line 209 - options include multi-agent therapy,  single-agent therapy ....

14. Line 307 - It is unclear from the author's description is SMAD4 is currently clinically used in RFA decision making.

Chemotherapy

15. I would recommend moving this section up the manuscript.

16. Figure#2 - unclear what is the connection between the text and Figure#2.

17. Since this section does touch upon personalized/patient-tailored medicine, I would recommend expanding this section with explanations on the HR pathway.  Please note that BRCA2 gene DOES NOT code for the BRCA1 protein (line 559). I would also mention PALB2 mutations, and address the concept of 'BRCAness" with examples of ATM, ATR, and RAD50 mutations. 

18. Probably important to mention the POLO study and discuss its findings when mentioning PARPi therapy. 

19. Line 556 - MSI can also be appear without a relation to Lynch. Should mention PD1/PDL-1 inhibitors. As they are approved in this group for advanced cancers and are associated with significant response.

20. Line 577 - Should probably cite the CODEBREAK100 study and discuss the specific findings.

21. If the goal is to discuss innovations - would recommend adding a section discussing endoscopic therapy which can replace surgery in advanced cancer cases through the use of LAMS. These can be used for an endoscopic gastric bypass, or billiary bypass. Endoscopy celiac neurolysis is also employed and is quite effective.

Since the authors are casting a wide net and covering many different aspects of pancreatic cancer care. some rearrangement of the paragraphs and different sections is recommended to improve clarity and readability. Currently, it is a bit awkward and cumbersome. 

Author Response

Dear Editor,

Many thanks for giving us the opportunity to resubmit our paper “STATE OF THE ART AND UPCOMING INNOVATIONS IN PANCREATIC CANCER CARE: A STEP FORWARD TO PRECISION MEDICINE”.

We also wish to thank the reviewers for their time and effort. We went through their interesting suggestions that allowed us to improve our paper.

Please, find a point-by-point answer to all addressed issues:

I thank the authors for the opportunity to review their study, entitled: "PERSONALIZED MANAGEMENT OF PANCREATIC CANCER: A STEP FORWARD TO PRECISION MEDICINE"

Overall, the manuscript is very expansive and covers many of the basic and faceted aspects of pancreatic cancer care.

The manuscript itself could benefit from some English language editing: There are no major grammatical errors overall, but the order and organization of the sentences and paragraph could be improved to increase readability and clarity.

Specific comments:

  1. Despite of the title, most of the manuscript does not touch upon personalized or precision medicine. In fact, I would estimate that only a page's worth discussed personalized medicine aspects. I would recommend to considers changing the title to something more fitting like "State of the art and upcoming innovations in pancreatic cancer care". R. We thank the reviewer for the interesting point of view. We modified the title as suggested.

Introduction:

  1. Consider citing more recent statistics (2023) for pancreatic cancer survival. R: We thank the reviewer for the comment and cited a recent study (Siegel, RL, Miller, KD, Wagle, NS, Jemal, A. Cancer statistics, 2023. CA Cancer J Clin. 2023; 73( 1): 17- 48. doi:10.3322/caac.21763) as suggested.
  2. Line 59 - Should be a start of a new paragraph. R: We thank the reviewer for the comment and modified as request.
  3. Line 84 - Imaging TNM ? R: We thank the reviewer for the correction and modified the text.
  4. Line 91 - Patients should be classified as anatomically resectable, other conditions may preclude them from surgery. R: We thank the reviewer for the comment and modified as request.

Advances in Pancreatic Surgery:

  1. Line 113 - Surgery coupled with post/perioperative chemotherapy is the only way to cure... R: We thank the reviewer for the comment and modified as suggested. 
  2. Figure#1 - Font is too small, please enlarge. R: We thank the reviewer for the comment and modified as suggested. 
  3. Line 122 - R0 is not microscopic. R: We thank the reviewer for the correction and modified the text.
  4. Line 125 - R1 is not macroscopic. R: We thank the reviewer for the correction and modified the text.
  5. Line 130 - Recent large studies use 20 or even 35 cases/year as the threshold for high-volume centers. R: We thank the reviewer for pointing this data updated and modified the reference.
  6. Line 131 - Are the authors referring to minimally invasive surgery (laparoscopic/robotic)? if so, what are the RCTs or Meta-analyses supporting MIS having lower mortality? R: We thank the reviewer for the accurate observation. Consequently, we rectified the text in order to prevent misunderstandings.
  7. Lines 170-199 - It may be worthwhile to cite some of the contrasting literature, as there is no clear consensus about the superiority of MIS in pancreatic surgery. Especially, since not all tumors are amenable to MIS resection and therefore there is in inherent selection bias in some of the studies. R: We thank the reviewer for the comment and better explicate the possible bias of the cited studies and the absence of clear consensus on surgical pancreatic cancer management.

Locoregional Therapy in Unresectable Pancreatic Cancer:

  1. Line 209 - options include multi-agent therapy,  single-agent therapy .... R: We thank the reviewer for the correction and modified as suggested.
  2. Line 307 - It is unclear from the author's description is SMAD4 is currently clinically used in RFA decision making. R: We thank the reviewer for the comment and better explained that currently the analysis of SMAD 4 mutation in clinical practice for guiding pancreatic cancer ablative therapy has not yet been established.

Chemotherapy

  1. I would recommend moving this section up the manuscript. R. We thank the reviewer for the interesting point of view. However, this order was purposely provided in order to give a more fluid reading and a better understanding of the topic.
  2. Figure#2 - unclear what is the connection between the text and Figure#2. R: We thank the reviewer for the comment and provided a new disposition of the figure in order to ensure a more fluid reading of the paragraph.
  3. Since this section does touch upon personalized/patient-tailored medicine, I would recommend expanding this section with explanations on the HR pathway.  Please note that BRCA2 gene DOES NOT code for the BRCA1 protein (line 559). I would also mention PALB2 mutations, and address the concept of 'BRCAness" with examples of ATM, ATR, and RAD50 mutations. R: we thank the reviewer for the suggestion, we added in the paragraph “Chemotherapy” a specific part on the mutations the reviewer suggested.
  4. Probably important to mention the POLO study and discuss its findings when mentioning PARPi therapy. R: we thank the reviewer, we added the study in the manuscript.
  5. Line 556 - MSI can also be appear without a relation to Lynch. Should mention PD1/PDL-1 inhibitors. As they are approved in this group for advanced cancers and are associated with significant response. R: we thank the reviewer for the comment, we modified the text as requested.
  6. Line 577 - Should probably cite the CODEBREAK100 study and discuss the specific findings .

R: We thank the reviewer for the comment, we added the trial above mentioned in the “chemotherapy” section.

  1. If the goal is to discuss innovations - would recommend adding a section discussing endoscopic therapy which can replace surgery in advanced cancer cases through the use of LAMS. These can be used for an endoscopic gastric bypass, or billiary bypass. Endoscopy celiac neurolysis is also employed and is quite effective. R: We thank the reviewer for the comment. Therefore, we have expanded the discussion by describing the endoscopy celiac neurolysis.

Reviewer 3 Report

Pancreatic Ductal Adeno Carcinoma (PDAC) is the third most common cause of cancer deaths in the United States and accounts for over 95% of all pancreatic cancers. The combined 1- and 5-year survival rates for PDAC are very poor, at 25% and 9% respectively. Despite new knowledge of the molecular profile of pancreatic cancer and its precursor lesions, survival rates have improved very little over the last 30 years. Considerable research has been focused on identifying molecular events in pancreatic carcinogenesis and their correlation with clinicopathological variables of pancreatic tumors, as well as developing effective therapies to fight the PDAC.

The proposed article is excellent review of currently available therapeutic and surgical approaches to treat different stages of PDAC. It is applausable that authors recognize the importance of personalized and multidisciplinary approaches for best outcome of each clinical situation.

Very well done and deserves to be published in “Cancers” with minor revisions.

Some suggestions:

Please, improve the resolution of the Figure 1.

Some typos have been noticed:

R.84 – please, check the abbreviation  “[IMMAGINE TMN]”

R.117 – please, put full stop after  “… BR-PDAC “

R.233 and R234 – please, change “… tumor; In patients with…” to “… tumor; in patients with…” and “…regions; In patients with an …” to “… regions; in patients with an …”

R.243 – please, delete one of 71 from “… no abdominal scarring7171. ”

R.343 – please, change  “… to neoadjuvant Chemotherapy” to “… to neoadjuvant chemotherapy”

R.442 – please, change  “… and intratumoral agent” to “… and intratumoral delivery of chemotherapeutic agents”

R.496 – please, change  “… of US-guided fiducial placement” to “… of EUS-guided fiducial placement”

R.508 – please, change  “… image-Guided Radiation Therapy” to “… image-guided radiation therapy”

R.517 – please, change  “Figure 2. physiopatological mechanisms of chemotherapy.” to “Figure 2. Physiopatological mechanisms of chemotherapy.”

R.546 – please, change  “… cisplatin, Nab-paclitaxel…” to “… cisplatin, nab-paclitaxel…”

Thank you!

Author Response

Dear Editor,

Many thanks for giving us the opportunity to resubmit our paper “STATE OF THE ART AND UPCOMING INNOVATIONS IN PANCREATIC CANCER CARE: A STEP FORWARD TO PRECISION MEDICINE”.

We also wish to thank the reviewers for their time and effort. We went through their interesting suggestions that allowed us to improve our paper.

Please, find a point-by-point answer to all addressed issues:

Pancreatic Ductal Adeno Carcinoma (PDAC) is the third most common cause of cancer deaths in the United States and accounts for over 95% of all pancreatic cancers. The combined 1- and 5-year survival rates for PDAC are very poor, at 25% and 9% respectively. Despite new knowledge of the molecular profile of pancreatic cancer and its precursor lesions, survival rates have improved very little over the last 30 years. Considerable research has been focused on identifying molecular events in pancreatic carcinogenesis and their correlation with clinicopathological variables of pancreatic tumors, as well as developing effective therapies to fight the PDAC.

The proposed article is excellent review of currently available therapeutic and surgical approaches to treat different stages of PDAC. It is applausable that authors recognize the importance of personalized and multidisciplinary approaches for best outcome of each clinical situation.

Very well done and deserves to be published in “Cancers” with minor revisions.

Some suggestions:

  1. Please, improve the resolution of the Figure 1. We thank the reviewer for the feedback, and as a result, we enlarged the imagine size.

Some typos have been noticed:

  1. R.84 – please, check the abbreviation  “[IMMAGINE TMN]” R: We thank the reviewer for this comment and erased the abbreviation.
  2. R.117 – please, put full stop after  “… BR-PDAC “ R: We thank the reviewer for this comment and corrected as request.
  3. R.233 and R234 – please, change “… tumor; In patients with…” to “… tumor; in patients with…” and “…regions; In patients with an …” to “… regions; in patients with an …”. R: We thank the reviewer for this comment and corrected as request.
  4. R.243 – please, delete one of 71 from “… no abdominal scarring7171. ”. R: We thank the reviewer for this comment and corrected as request.
  5. R.343 – please, change  “… to neoadjuvant Chemotherapy” to “… to neoadjuvant chemotherapy”. R: We thank the reviewer for this comment and corrected as request.
  6. R.442 – please, change  “… and intratumoral agent” to “… and intratumoral delivery of chemotherapeutic agents”. R: We thank the reviewer for this comment and modified as request.
  7. R.496 – please, change  “… of US-guided fiducial placement” to “… of EUS-guided fiducial placement”. R: We thank the reviewer for this comment and modified as request.
  8. R.508 – please, change  “… image-Guided Radiation Therapy” to “… image-guided radiation therapy”. R: We thank the reviewer for this comment and modified as request.
  9. R.517 – please, change  “Figure 2. physiopatological mechanisms of chemotherapy.” to “Figure 2. Physiopatological mechanisms of chemotherapy.” R: We thank the reviewer for this comment and modified as request.
  10. R.546 – please, change  “… cisplatin, Nab-paclitaxel…” to “… cisplatin, nab-paclitaxel…” R: We thank the reviewer for this comment and modified as request.

The following points required explanation and revision: 

  1. in subchapter 3. Acute pancreatitis, one sentence is repeated, but there is another citation at the end - which is the correct one? maybe both? (9, 10). R: We thank the reviewer for having pointed out this discrepancy. The correction was made with the insertion of the appropriate citation.

  1. in subchapter 4. Clinical Nutrition, the abbreviation FFMI has not been explained.

R: We thank the reviewer for this comment. The abbreviation "FFMI" has now explicated in its full name: “Fat Free Body Mass Index”.

  1. in subchapter 12. Modulation of Microbiota in Acute Pancreatitis there are sentences that in the meta-analysis included a total of 3.864 SAP patients the administration of carbapenem as antibiotic prophylaxis in SAP was associated with a statistically significant reduction of complications and infections. The next sentence: Routinary use of carbapenem antibiotics is not recommended (item 173 in references). The authors should explain why they are not recommended despite the reduction of complications and infections.

R: We thank the reviewer for the comment. As suggested, we modified subchapter 12, to better explain why routine use of carbapenem antibiotics is not recommended despite the reduction in infections and complication.

Round 2

Reviewer 2 Report

I thank the authors for their revised manuscript.

I have only a few minor comments that once addressed will allow acceptance in my view:

Chemotherapy:

1. "Recent studies in patients with mutations in the breast cancer susceptibility proteins (BRCA1/2 genes)...."

2. I'm assuming "PARPB gene" = PALB2 gene

3. Data from CODEBREAK 100 is already available, see “Strickler, et al. Sotorasib in KRAS p.G12C–Mutated Advanced Pancreatic Cancer. NEJM. 2023” which already included survival data for metastatic patients treated with sotorasib.

I would still recommend considering some minor language editing for clarity and readability as some sentences have multiple compound statements which slow down the flow of reading.

Author Response

Dear Editor,

Many thanks for giving us the opportunity to resubmit our paper “STATE OF THE ART AND UPCOMING INNOVATIONS IN PANCREATIC CANCER CARE: A STEP FORWARD TO PRECISION MEDICINE”

We also wish to thank the reviewers for their time and effort.

Please, find a point-by-point answer to all addressed issues:

Chemotherapy:

  1. "Recent studies in patients with mutations in the breast cancer susceptibility proteins(BRCA1/2 genes)...." R: we thank the reviewer for the correction, we modified the text as indicated.
  2. I'm assuming "PARPB gene" = PALB2 gene. R: we thank the reviewer for the correction, we modified the text as suggested.
  3. Data from CODEBREAK 100 is already available, see “Strickler, et al. Sotorasib in KRAS p.G12C–Mutated Advanced Pancreatic Cancer. NEJM. 2023” which already included survival data for metastatic patients treated with sotorasib. R: we thank the reviewer for the comment, we integrated the suggested article.

Finally, we edited some sentences in the text in order to enhance the text’s fluency and improve its clarity of comprehension.